# Comparison of 24-h Diet Records, 24-h Urine, and Duplicate Diets for Estimating Dietary Intakes of Potassium, Sodium, and Iodine in Children

**DOI:** 10.3390/nu11122927

**Published:** 2019-12-03

**Authors:** Rana Peniamina, Sheila Skeaff, Jillian J. Haszard, Rachael McLean

**Affiliations:** 1Department of Preventive and Social Medicine, University of Otago, Dunedin 9016, New Zealand; rana.peniamina@otago.ac.nz; 2Department of Human Nutrition, University of Otago, Dunedin 9016, New Zealand; sheila.skeaff@otago.ac.nz (S.S.); jill.haszard@otago.ac.nz (J.J.H.)

**Keywords:** potassium, sodium, iodine, children, dietary intakes, 24-h urine, diet records, duplicate diets

## Abstract

Accurately estimating nutrient intake can be challenging, yet it is important for informing policy. This cross-sectional validation study compared the use of three methods for estimating the intake of sodium, potassium, and iodine in children aged 9–11 years in New Zealand. Over the same 24 hour period, participants collected duplicate diets (*n* = 37), weighed food records (*n* = 84), and 24 hour urine samples (*n* = 82). Important differences were found between dietary estimates of sodium, potassium, and iodine using the three methods of dietary assessment, suggesting that different methods of assessment have specific limitations for the measurement of these nutrients in children. Bland Altman plots show relatively wide limits of agreement for all measures and nutrients. These results support the World Health Organization’s (WHOs) recommendations to use urinary assessment to measure population sodium and iodine intake, while dietary assessment appears to be more accurate for estimating potassium intake. Compared to reference values, our results suggest that the children in this study consume inadequate iodine, inadequate potassium, and excess dietary sodium. Public health measures to reduce sodium intake, increase intake of fruit and vegetables, and iodine-rich foods are warranted in New Zealand.

## 1. Introduction

Dietary intakes of sodium and potassium have important consequences for the health of both adults and children. A high intake of dietary sodium is associated with negative health outcomes such as increased blood pressure [1,2], and an increased risk of cardiovascular disease [3,4,5,6], gastric cancer [7,8,9], and renal disease [10,11]. Dietary potassium, a marker of fruit and vegetable intake, is associated with health benefits such as blood pressure-lowering [1], and a reduced risk for stroke [3]. Urinary potassium has also been linked to better bone-mineral density in prepubertal children [12]. In addition, a high dietary intake of potassium may mitigate the negative consequences of high sodium consumption. A lower ratio of sodium to potassium has been linked to a reduced risk for obesity, cardiovascular disease, and chronic kidney disease [4,10,13,14,15,16].

With the hope of reducing the burden of noncommunicable diseases in the population, the World Health Organization recommends reducing dietary intake of sodium and increasing dietary intake of potassium [17,18]. Concurrent monitoring of iodine and sodium intakes is important to ensure reductions in sodium intake are not adversely affecting iodine intake. If sodium intake reduces over time, iodine intake can be maintained through increasing the concentration of iodine in table salt, mandating the use of iodized salt in processed foods, or fortifying other foods with iodine. Monitoring of dietary sodium and potassium intakes at a population level is necessary for understanding whether recommended levels are being achieved and whether efforts to promote healthy eating are successful. 

Estimating the intakes of sodium, potassium, and iodine can, however, be challenging. While a number of dietary assessment methods are available, there is a trade-off between the validity of the results versus increased expense and participant burden. Less expensive, low participant burden methods (e.g., food frequency questionnaires, 24-h diet recall) are associated with methodological challenges such as the tendency for participants to under-report intakes (particularly for foods considered to be unhealthy and for snack foods), recall bias, difficulty quantifying discretionary salt, and the variability of sodium concentration in foods within a food category. The accuracy of estimates also depends on whether the food composition database used is relevant to the local food supply, up-to-date, and comprehensive. Duplicate diets, weighed food records, and biomarkers in 24-h urine samples are examples of more accurate methods for determining dietary nutrients. However, duplicate diets involve participants collecting a duplicate portion of all food and drink consumed over a specified period for subsequent laboratory analysis, which is time-consuming, expensive, and requires considerable effort by participants. Collection of 24-h urine is also associated with higher participant burden, and the accuracy of the results can be limited by under or over- collection. In addition, 24-h urine samples are not suitable for some population groups such as very young children (where complete 24-h urine samples are difficult to achieve), and accuracy can be affected by factors such as health status, hydration status, excessive sweating, hormonal changes, and acute dietary transition [19]. 

The current research compared the use of three methods (24-h diet records, duplicate diets, and 24-h urine) for estimating the intake of sodium, potassium, and iodine in children aged 9–11 years in New Zealand. Accurate monitoring of population nutrient intakes into the future is important for informing policy. Key policy areas include dietary sodium reduction and iodine fortification of discretionary salt. New Zealand and Australian Nutrient Reference Values [20] recommend adequate intakes (AI) for iodine of 120 mcg/day for both girls and boys and for potassium of 2500 mg/day for girls and 3000 mg/day for boys in this age group, respectively. The recommended Upper Level (UL) for sodium intake in this age group is 2000 mg/day. The primary aim of this study was to measure the agreement between nutrients in 24-h diet records, duplicate diets, and 24-h urine, and to determine the degree of bias in 24-h diet record assessment in children. The intakes of these nutrients were also compared to the nutrient reference values for this age group.

## 2. Materials and Methods

### 2.1. Study Design and Recruitment

This cross-sectional study was conducted in the Otago region in the South Island of New Zealand, with data collection taking place during March–May 2017 and March–May 2018. Participants for this research were healthy children aged 9 to 11 years at the time of the study. Siblings were excluded. Children were recruited via word of mouth, through local primary schools, and by advertising on Facebook. Researchers contacted the schools via email and asked them to distribute the study information pack to children of suitable ages. Interested children and caregivers were provided with a participant information form and consent form. Children and caregivers who returned a signed consent form were contacted via telephone or email to arrange a home visit by the researchers for the start of data collection. At the first home visit, the caregivers of each child were provided with a form for completing the 24-h diet record, electronic kitchen scales, a booklet with instructions for collecting the duplicate diet, a 2L bucket with lid for food sample collection, a 2L bottle for 24-h urine samples, and a funnel to aid the collection of urine. Verbal instructions detailing how and when to collect samples were also given, and the importance of the children following their usual diet and drinking habits was emphasized. At a subsequent home visit on the day after sample collection, the researchers collected the completed 24-h diet record, the food vessel containing the duplicate diet, and the bottle containing the 24-h urine sample. The study was approved by the University of Otago Health Ethics Committee (HE17/001). 

### 2.2. Demographic Data

Each child completed an online questionnaire collecting information about the child’s age, date of birth, sex, ethnicity, school, and use of vitamin/mineral supplements. The child’s weight was measured by the researchers using a Seca electronic scale, then entered into the online questionnaire.

### 2.3. 24-h Diet Records and Duplicate Diets

The 24-h diet records were completed with details about the food or drink consumed (including the brand, preparation/cooking method, and weight) at the time of consumption. For recipes, weights of individual ingredients and the total recipe were recorded as well as the weight of the portion of the recipe eaten by the child. Each diet record was entered into Kaiculator (a food analysis program developed at the University of Otago) and analyzed for daily sodium, potassium, and iodine.

Duplicate diets were prepared by caregivers observing what the child ate over a period of 24 h and placing a replicate version in the 2L collection bucket. Uneaten parts of meals (including bones, fruit/vegetable cores, and skins) were not included in the duplicate diet bucket. The duplicate diet was stored in a refrigerator until collection by the researchers. Collected duplicate diets were weighed to determine the total weight of food eaten by the child. The food was then homogenized until smooth, using a Waring blender. Remaining food residues were rinsed from the collection bucket into the blender using known amounts of deionized water. Homogenized samples were frozen at −20 °C until required for analysis.

For sodium and potassium analysis, duplicate diet samples were dry-ashed at 450 °C in a muffle furnace [21]. Ashed samples were dissolved in HCl, transferred into acid-washed 100 mL volumetric flasks, and then adjusted to volume with distilled, deionized water. The sodium and potassium concentration of the solutions were analyzed by flame atomic absorption spectrophotometry (AAS) (AA-800, Perkin-Elmer Corp., Norwalk, CT, USA). For iodine analysis, samples were sent out to Hill Laboratories (Hamilton, New Zealand). The concentration of iodine in the samples was determined using Tetramethylammonium hydroxide micro digestion and analysis by inductively coupled plasma mass spectrometry (ICP-MS), as described by Fecher et al. [22].

### 2.4. Urine Sample and Urinary Sodium, Potassium, and Iodine Analysis

The children collected their urine over the same 24-h time period as their weighed diet record and duplicate diet. The time of the first void of the day was recorded by the caregivers, and this urine was discarded. All urine for the remaining 24-h (including the first void of the following morning) was collected into a 2L screw-top bottle with the aid of a funnel. Any missing void was noted. The total volume (weight) of urine was recorded by the researchers. Completeness of the 24-h urine samples was assessed by comparing them to the lower limit for children older than 6 years (9 mL/h) [23]. Samples that did not meet this criterion were excluded from analysis, as were samples where participants recorded missing more than one void in the 24-h period. Urine samples were analyzed for sodium, potassium, and iodine using inductively coupled plasma mass spectrometry (ICP-MS) (Agilent 7500ce ICP-MS, Agilent Technologies, Tokyo, Japan).

### 2.5. Statistical Analysis

All statistical analysis was carried out with Stata 15.1 (StataCorp, College Station, TX, USA). Intakes of sodium, potassium, and iodine were described using means and standard deviations (SD) of the data that had been adjusted for the usual intake. This adjustment takes participants with repeat measures and estimates the within-person variation, which was then used to adjust all values. The Multiple Source Method (MSM) program was used to undertake this adjustment [24]. Within-person variation was reported using within-person means and SD and is presented in the Appendix A. 

Agreement between the 3 methods of assessment was investigated using the data from the first day of measurement (i.e., repeats were not used). Mean differences and 95% confidence intervals were calculated between methods with a paired t-test. Both Spearman’s and intraclass correlation coefficients were calculated to assess how strongly the methods were related to each other. Bland–Altman plots were generated, displaying the mean difference and the 95% limits of agreement (LOA).

## 3. Results

Table 1 shows the characteristics of the study participants. The participants ranged from 9 to 11 years in age, 48.8% were female, and most were of New Zealand European and other (NZEO) ethnicity (86.9%). The school decile (an indicator of the socio-economic status of the school population) of the participants’ schools ranged from 3 to 10.

Mean measurements of sodium, potassium, and iodine in the study population using the three different methods (24-h urine, 24-h diet record, and duplicate diet) are shown in Table 2, along with the mean differences between the methods and 95% confidence intervals. Mean intakes of sodium, potassium, and iodine by sex and method of measurement are presented in Appendix A. The data suggested that, on average, the children in this study were consuming diets with higher than recommended levels of sodium, and lower than recommended levels of potassium and iodine. The 24-h record and the 24-h urine gave similar results for sodium intake, with the duplicate diet measuring sodium intakes to be considerably lower. In contrast, the 24-h urine measured potassium intakes to be much lower than the 24-h record and duplicate diet. The duplicate diet data yielded higher iodine estimates compared to the 24-h urine and 24-h diet record methods, with the 24-h record giving the lowest estimates. (means of 74 μg and 52 μg, respectively). 

Bland–Altman plots (Figure 1, Figure 2 and Figure 3) illustrate reasonably consistent agreement between different methods of measurement, however, the limits of agreement were wide, suggesting poor accuracy. The pattern of variability between the measures appeared to differ for the three nutrients. Intra-class correlations were strongest between the 24-h recall and the duplicate diet for all three nutrients (Appendix A).

## 4. Discussion

There were important differences found between dietary estimates of sodium, potassium, and iodine using three methods of dietary assessment: Duplicate diets, weighed diet records, and 24-h urine in children aged 9–11 years. In this study, the three assessment methods were undertaken in the same children for the same 24-h period, thus these differences cannot be explained by day to day variability in nutrient intakes. There appeared to be no consistent pattern with respect to differences between the three different methods of assessment across the three nutrients, which suggests that different methods of assessment have specific limitations for the measurement of sodium, potassium, and iodine in children. 

For sodium, the duplicate diet appeared to underestimate intake compared to both the 24-h urine and 24-h diet record, which give similar estimates. For potassium, the 24-h urine appeared to underestimate intake compared to the dietary assessment methods (food record and duplicate diets), and for iodine, the three measures are similar, although food records appear to underestimate intake compared to the other two measures. That these patterns are different across the three nutrients suggests no inherent bias associated with one method of assessment, such as under or overcollection of urine, duplicate diet, or dietary information. The Bland–Altman plots show relatively wide limits of agreement for all measures and nutrients. Patterns of agreement are wider at higher levels of intake when comparing diet records and 24-h urine (Figure 1). This is consistent with some studies in adults where underreporting of dietary intake is associated with higher body mass index (and presumably higher usual energy intake), resulting in greater differences in estimates at higher intakes [25,26]. There is no obvious systematic bias indicated by the plots, which include duplicate diets, although this may be partly explained by the smaller numbers of participants in these plots. (Figure 2 and Figure 3). 

Our results indicate that there is no systematic bias in a single dietary assessment method, but rather different methods may be more accurate for different nutrients of interest. Although duplicate diets theoretically contain a complete record of all foods consumed during the period, other research has shown that this method can underestimate intake compared with other methods of assessment [27,28,29]. Trijsberg et al. collected two days of duplicate diets and compared findings with two 24-h urine samples on different days in adults and found under-estimation of usual intake of sodium and potassium on the days where duplicate diets were collected. They speculated that duplicate diet collection may have resulted in lower consumption on those days [27]. Kim et al. collected multiple duplicate diets, food records, and 24-h urines over a 12-month period in 29 healthy volunteers aged 20–53 and found lower energy intake overall when duplicate diet collections were being made, compared with diet records and 24-h urine samples. Interestingly, this effect was stronger in younger participants than older participants [28]. Our results show that the effect may not be due to a decrease in intake when collecting duplicate diets but an under-collection of food at this time, at least in children. Further, our results suggest that some foods may be less likely to be included than others in this age group. In our study, duplicate diets underestimated sodium intake but not potassium intake when compared to the other two methods, suggesting a systematic measurement error. This may indicate that salty foods such as snacks and discretionary salt have not been recorded, but fruit and vegetables were included or even over-estimated compared with levels indicated by 24-h urine samples. The extent to which this is affected by social desirability bias or recall bias (salty snacks consumed between meals are more likely to be forgotten than vegetables included in a meal for example) is not clear. 

For potassium, urinary excretion underestimated intake compared with the dietary assessment measures. Although both sodium and potassium intakes are often measured using 24-h urine, previous evidence suggests that urinary excretion of potassium may under-estimate intake more than urinary sodium. In adults, an average of 90% of ingested dietary sodium is excreted in the urine over a 24-h period, although this proportion may vary slightly from day to day [19,30]. Controlled feeding studies in adults suggest that a lower proportion of dietary potassium is excreted in the urine. Participants in the Dietary Approaches to Stop Hypertension (DASH) trial excreted between 50%–74% of ingested potassium in the urine over a 24-h period, with black participants having lower excretion proportions than white [31]. The Mars studies (Mars 105 and Mars 520) tightly controlled dietary intake in 4 and 6 healthy men (respectively) over 105 (Mars 105) and 205 (Mars 520) days, while temperature and humidity were kept constant for an extended period. Multiple twenty four hour urinary specimens were collected and showed mean urinary potassium recovery of 99.8% (SD 23.0) for Mars105, and 77.9% (SD 24.3) for Mars520 [32]. Therefore, the true intake of potassium in the children in this study may lie somewhere between the estimates from urinary excretion and dietary measures given the potential for under excretion and measurement error in both measures. For iodine, intake estimates from 24-h urine and duplicate diet are most similar, suggesting that 24-h diet record may not be an appropriate measure. Our study supports WHO and other current recommendations to measure population iodine intake in children using urinary assessment [33,34]. These results also support WHO recommendations to use 24-h urine to assess population sodium intake [35], although further dietary assessment (in this case weighed diet record) appears to be more accurate than urinary assessment for estimating potassium intake. Further research involving controlled feeding methods would help clarify the extent to which urinary assessment accurately reflects potassium intake in children of this age group. 

Compared to reference values, our mean results suggest that the children in this study consume inadequate iodine, inadequate potassium, and excess dietary sodium. Across all three measures of the three nutrients, only one measure is consistent with nutrient reference values (1886 mg sodium/day for duplicate diets), but as discussed above, we believe this is likely to be an under-estimate. Strategies to reduce sodium intake, increase intakes of fruit, vegetables and iodine-rich foods are warranted. 

A strength of this study is the measurement of three nutrients using three different assessment methods over the same 24-h period. The sample includes children of varying socioeconomic status and important ethnic groups in New Zealand. However, it is not representative of the New Zealand population as Māori and Pacific children were under-represented compared to the total population, and children were only recruited from the lower South Island of New Zealand. Although every effort was made to enable and support the children and their families to collect complete samples and information, measurement error cannot be excluded as this study was not conducted in a controlled environment. 

## 5. Conclusions

This study highlights the difficulty of collecting accurate dietary information in free-living children. Although 24-h urine appears to measure intake of sodium and iodine adequately, it appears to substantially underestimate potassium intake, at least compared to dietary assessment (24-h diet record and duplicate diets). Further research in tightly controlled feeding studies would elucidate the extent to which urinary assessment accurately represents intake over a 24-h period. However, the practicalities associated with keeping children in a confined environment and range of energy requirements in this age group would make such a study challenging. Regardless, these children appear not to be meeting the New Zealand and Australian dietary recommendations for sodium, potassium, and iodine intake. A national children’s nutrition survey is needed to confirm these results, and public health measures to reduce sodium intake, and increase intake of fruit, vegetables and iodine-rich foods are warranted. 

## Figures and Tables

**Figure 1 nutrients-11-02927-f001:**
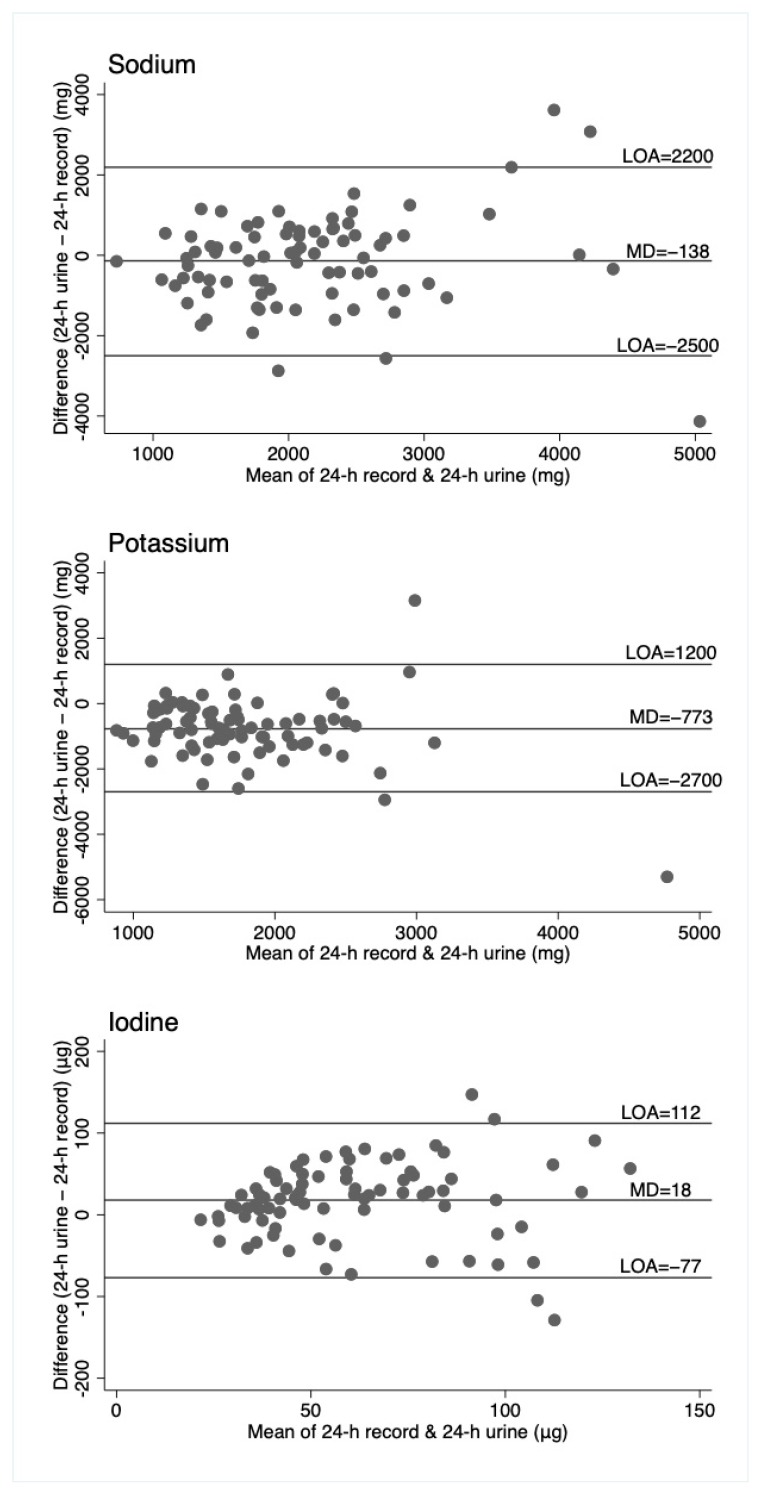
Bland–Altman plots for sodium, potassium, and iodine, comparing 24-h diet records to 24-h urine. Horizontal lines represent the mean difference (MD) and the 95% limits of agreement (LOA). N = 82.

**Figure 2 nutrients-11-02927-f002:**
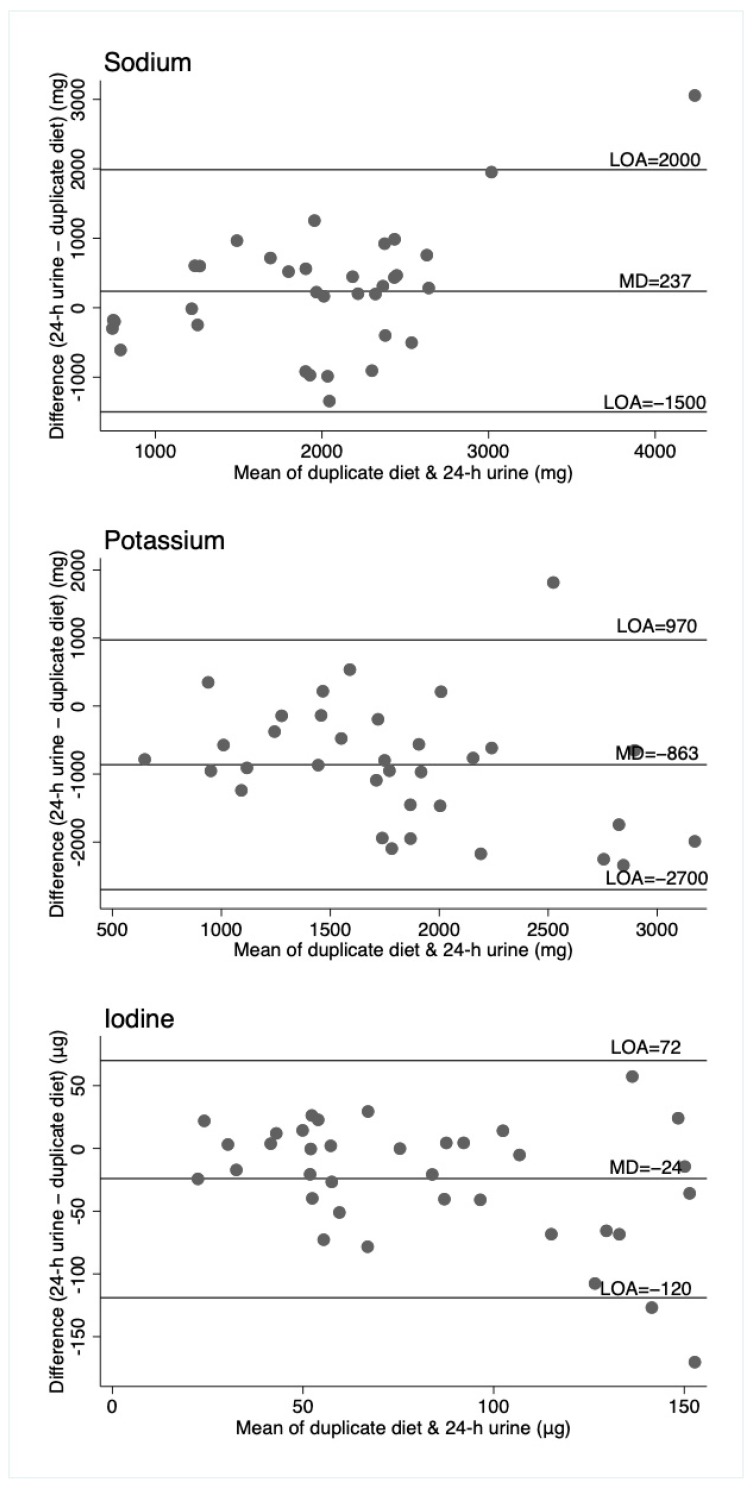
Bland–Altman plots for sodium, potassium, and iodine, comparing duplicate diet to 24-h urine. Horizontal lines represent the mean difference (MD) and the 95% limits of agreement (LOA). N = 36 (*n* = 34 for sodium and potassium).

**Figure 3 nutrients-11-02927-f003:**
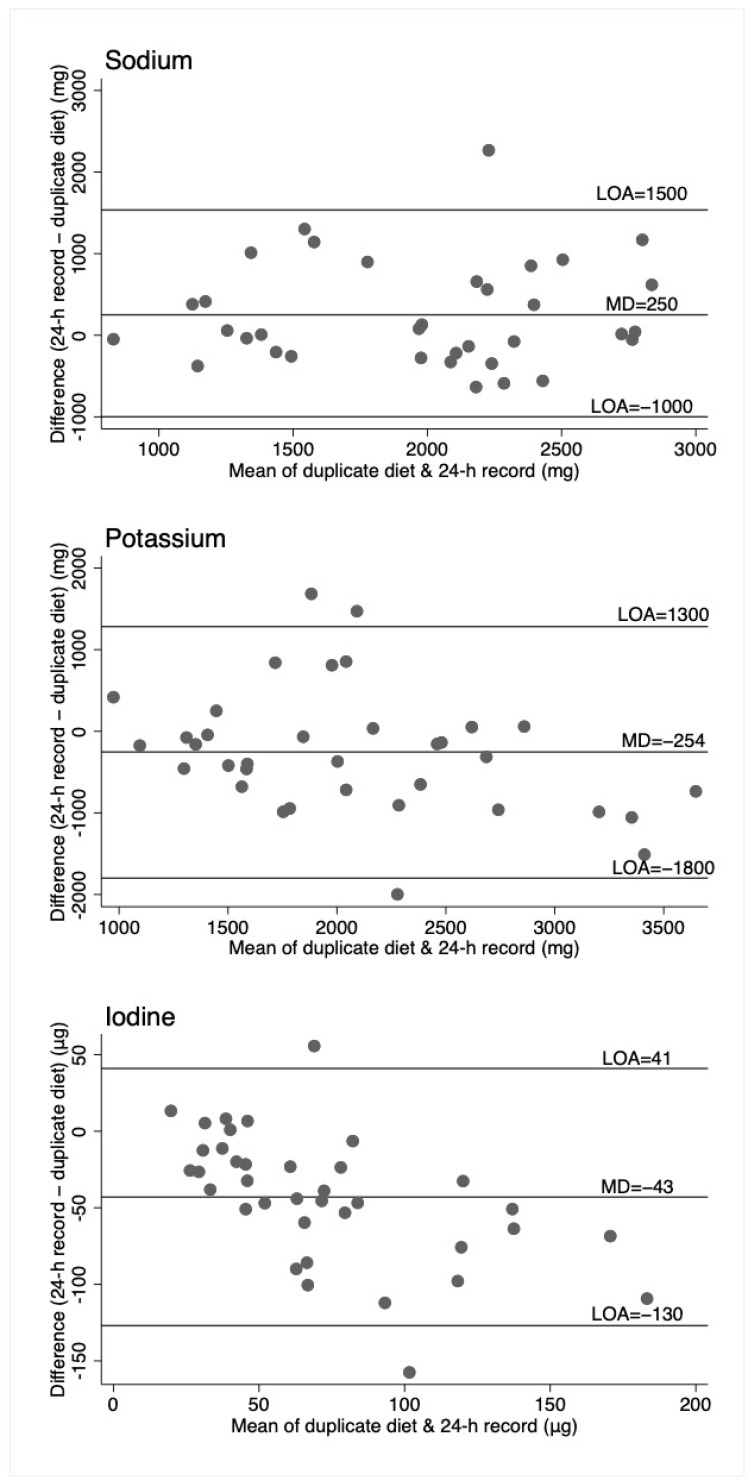
Bland–Altman plots for sodium, potassium, and iodine, comparing the duplicate diet to 24-h diet record. Horizontal lines represent the mean difference (MD) and the 95% limits of agreement (LOA). N = 37 (*n* = 35 for sodium and potassium).

**Table 1 nutrients-11-02927-t001:** Participant characteristics (*n* = 84): mean (standard deviation) unless otherwise specified.

Characteristic	N (%) ^a^
Age, years	9.6 (0.7)
Sex, female	41 (48.8)
Ethnicity	
NZEO ^b^	73 (86.9)
Māori or Pacific	11 (13.1)
School decile, median (25th, 75th percentile)	8 (6, 9)

^a^ Unless otherwise specified; ^b^ New Zealand European and Other (this included participants who identified with other ethnicities: Asian *n* = 2, Middle Eastern/Latin American/ African *n* = 1).

**Table 2 nutrients-11-02927-t002:** Differences between measurements of sodium, potassium, and iodine intakes (total *n* = 84).

	N ^b^	Sodium (mg)	Potassium (mg)	Iodine (μg)
24-h urine ^a,b^, mean (SD)	82	2119 (439)	1414 (345)	74 (17)
24-h record ^a^, mean (SD)	84	2223 (152)	2108 (111)	52 (6)
Duplicate diet ^a,b^, mean (SD)	37	1886 (405) ^b^	2172 (593) ^b^	95 (18)
Mean differences ^a^ (95% CI)				
24-h record -24-h urine	82	138 (−118, 394)	773 (557, 990)	−18 (−28, −7)
duplicate diet -24-h urine	36	−237 (−542, 69) ^b^	863 (543, 1183) ^b^	24 (8, 40)
duplicate diet -24-h record	37	−250 (−471, −29) ^b^	254 (−10, 519) ^b^	43 (29, 57)
Mean differences (95%CI) for participants with all three measures ^a^				
24-h record -24-h urine	34	22 (−363, 407)	606 (320, 902)	−20 (−35, −5)
duplicate diet -24-h urine	34	−237 (−542, 69)	863 (543, 1183)	24 (8, 40)
duplicate diet -24-h record	34	−259 (−486, −32)	257 (−16, 549)	44 (29, 58)

^a^ Measures adjusted for usual intake (using a subsample of repeats) were used to describe mean intakes; measures from the first day (i.e., not the repeat days) were used to describe the differences between measurement methods; ^b^ Two participants were excluded from urine analysis due to incomplete sample, 37 participants agreed to duplicate diet, but two of these were not able to be calculated for sodium and potassium.

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
