# Peer review of "Comparison of 24-h Diet Records, 24-h Urine, and Duplicate Diets for Estimating Dietary Intakes of Potassium, Sodium, and Iodine in Children"

_nutrients, 2019, doi:10.3390/nu11122927_

Round 1
Reviewer 1 Report
In this study, Peniamina et al. compared the use of three methods -24-h diet records, duplicate diets, and 24-h urine- for estimating the intake of Sodium, Potassium, and Iodine in children 9-11 years old in New Zealand. They showed that 24-h diet records and 24-h urine yielded similar results for sodium intake, with the duplicate diet showing lower sodium intakes. For potassium, 24-h diet record and duplicate diet yielded similar results. The 24-h diet record measured potassium intakes to be much lower. The duplicate diet yielded the highest estimate for Iodine intake followed by 24-h urine. The 24-h diet record measured iodine intakes to be much lower. They concluded that there is no systematic bias in a single dietary assessment method, but rather different methods may be more suitable for estimating consumption of a given nutrient. In addition, based on their results, they report that children ages 9-11 in New Zealand do not consume the recommended daily amounts for these nutrients.
Overall, the study was scientifically sound and well conducted, and the manuscript is well written. The authors did a great job identifying the limitations of their study. The article can, however, benefit from several improvements.
Major comments:
-Are there any statistically significant differences in the estimates obtained for each nutrient between the three methods tested? Providing this information would significantly improve the quality of the manuscript.
-Table 2: Since the recommended daily amounts are different for boys and girls, I’d suggest presenting the data for boys and girls separately.
Minor comments:
-There is a typo in Table 2: column 3, row 11.
-The authors don’t mention the recommended daily values for Sodium, Potassium, and Iodine until the conclusion section. I’d suggest providing this information in the introduction as it would help the reader put the results presented in the manuscript into perspective.
Reviewer 2 Report
I only have a few comments/suggestions.
Lines 39-47: Sodium reduction and iodine fortification are not in conflict; if sodium consumption is reduced, the fortificant level can be increased to achieve the same iodine intake. This should be included in the paper so that readers do not misinterpret the relationship between the two public health interventions. Furthermore, it makes the paper more relevant because if sodium intake is changing, it needs to be monitored regularly to adjust iodine fortification.
Line 111, 218, 237, 279: typos (ample, missing “to”, X, missing “survey”
Line 258: results cannot be generalized to “children in NZ”. Suggest rewording. Also, was there a change over time in lower South Island? A shift in diet with public health implications would help justify the recommendation for a new national survey. Otherwise, the recommendation is not really related to the study findings.
